# Capsaicin and Its Effect on Exercise Performance, Fatigue and Inflammation after Exercise

**DOI:** 10.3390/nu14020232

**Published:** 2022-01-06

**Authors:** Gaia Giuriato, Massimo Venturelli, Alexs Matias, Edgard M. K. V. K. Soares, Jessica Gaetgens, Kimberley A. Frederick, Stephen J. Ives

**Affiliations:** 1Department of Neurosciences, Biomedicine and Movement Sciences, University of Verona, 37134 Verona, Italy; gaia-giuriato@univr.it (G.G.); Massimo.venturelli@univr.it (M.V.); 2Health and Human Physiological Sciences Department, Skidmore College, Saratoga Springs, NY 12866, USA; amatias@skidmore.edu (A.M.); edgardsoares@gmail.com (E.M.K.V.K.S.); 3Department of Internal Medicine, University of Utah, Salt Lake City, UT 84132, USA; 4Study Group on Exercise and Physical Activity Physiology and Epidemiology, Exercise Physiology Laboratory, Faculty of Physical Education, University of Brasilia—UnB, Brasilia 70910-900, Brazil; 5Department of Chemistry, Skidmore College, Saratoga Springs, NY 12866, USA; jgaetgens@skidmore.edu (J.G.); kfreder1@skidmore.edu (K.A.F.)

**Keywords:** motoneuron, afferent, skeletal muscle, cardiac output, ventilation, metabolism, perfusion

## Abstract

Capsaicin (CAP) activates the transient receptor potential vanilloid 1 (TRPV_1_) channel on sensory neurons, improving ATP production, vascular function, fatigue resistance, and thus exercise performance. However, the underlying mechanisms of CAP-induced ergogenic effects and fatigue-resistance, remain elusive. To evaluate the potential anti-fatigue effects of CAP, 10 young healthy males performed constant-load cycling exercise time to exhaustion (TTE) trials (85% maximal work rate) after ingestion of placebo (PL; fiber) or CAP capsules in a blinded, counterbalanced, crossover design, while cardiorespiratory responses were monitored. Fatigue was assessed with the interpolated twitch technique, pre-post exercise, during isometric maximal voluntary contractions (MVC). No significant differences (*p* > 0.05) were detected in cardiorespiratory responses and self-reported fatigue (RPE scale) during the time trial or in TTE (375 ± 26 and 327 ± 36 s, respectively). CAP attenuated the reduction in potentiated twitch (PL: −52 ± 6 vs. CAP: −42 ± 11%, *p* = 0.037), and tended to attenuate the decline in maximal relaxation rate (PL: −47 ± 33 vs. CAP: −29 ± 68%, *p* = 0.057), but not maximal rate of force development, MVC, or voluntary muscle activation. Thus, CAP might attenuate neuromuscular fatigue through alterations in afferent signaling or neuromuscular relaxation kinetics, perhaps mediated via the sarco-endoplasmic reticulum Ca^2+^ ATPase (SERCA) pumps, thereby increasing the rate of Ca^2+^ reuptake and relaxation.

## 1. Introduction

The primary pungent bioactive ingredient in hot peppers, capsaicin (CAP), has long been regarded for its therapeutic potential. Capsaicin (8-methyl-N-vanillyl-trans-6-nonenamide) is classically described as an irritant and is a well-known endogenous activator of the transient receptor potential vanilloid type 1 (TRPV_1_) on sensory neurons modulating signals for heat and/or pain. Exposure to CAP triggers a potent neuronal calcium influx, often followed by a reflex down-regulation of the TRPV_1_ activity [1,2,3]. For this reason, CAP is a promising clinical tool to modulate TRPV_1_-related pathways, from pain perception [1,2,3,4], inflammation [5], and immunity [6], to most severe pathologies like schizophrenia [7], anxiety, depression [8], obesity [9] and chronic fatigue [10]. Ingestion of CAP increases thermogenesis by stimulating catecholamine secretion from the adrenal medulla, decreasing adipogenesis, and enhancing energy metabolism [11,12,13,14,15], improving mitochondrial biogenesis and adenosine triphosphate (ATP) synthesis, and is even suggested to improve markers of cardiovascular health [16,17,18,19,20]. 

In rodents, CAP elicits a spontaneous active behavior, increases grip strength and swimming time to exhaustion in a dose-dependent manner [21,22,23,24]. These enhancements in physical performance were correlated to increases in the hepatic glycogen content [21], likely as a result of glycogen sparing [24] and elevated fatty acid utilization due to CAP-induced adrenal catecholamine secretion [22]. Moreover, studies on mice showed that the TRPV_1_ activation by CAP administration upregulates PGC-1α, promotes mitochondrial biogenesis, increases the contribution of oxidative ATP production, and upregulates the expression of oxidative fibers in skeletal muscle [25,26]. In a murine model, CAP-induced muscular relaxation is mediated via a direct inhibitory action on the voltage-operated Ca^2+^ channels inside the cell [4]. In addition, a single high dose of CAP downregulates the expression of the mitochondrial uncoupling protein UCP3, and reduces the ATP cost of contraction, despite an unchanged, and at times increased, electrical twitch force generation [25,27]. Although CAP has been widely studied in cell and murine models, its acute in vivo physiological effects when combined with exercise have received relatively minimal attention, especially in humans.

Researchers have explored the effects of CAP ingestion and its influence on different exercise paradigms in healthy males [28,29,30,31]. Thus there have been some reports of performance improvements induced by the consumption of a single 12 mg dose of purified CAP during a 1500-m running time trial [30], high-intensity intermittent exercise [28], and resistance training [29], but not during a 10 km running performance [31]. Additionally, CAP reduced the rating of perceived exertion (RPE) during the endurance and resistance tasks, with no between-group differences in lactate concentration, suggesting a possible mediating effect of CAP on fatigue or sensations of fatigue. On the contrary, Opheim and colleagues did not observe any effect of 7 days of ingestion of 28.5 mg of CAP on performance or level of perceived fatigue during repeated sprint intervals (15 × 30 m sprints with intervals of 35 s), but this dosing regimen induced significant gastrointestinal distress [32], highlighting the importance of dosage. Furthermore, these aforementioned studies on CAP focused solely on exercise performance, leaving the underlying mechanisms of CAP on the fatiguing process largely unexplored.

Exercise increases circulating concentrations of specific inflammatory cytokines, e.g., interleukin-6 (IL-6) and interleukin-1β (IL-1β) [33,34], which have been suggested as potential mediators of central nervous system fatigue in different diseases [35]. High-intensity exercise also increases salivary α-amylase activity [36] and cortisol levels [37], which likely reflect the neuroendocrine response to exercise; cortisol has been shown to have anti-inflammatory properties, so the inflammatory and anti-inflammatory responses should be considered together. Furthermore, CAP has known analgesic and anti-inflammatory properties, along with the capacity to reduce the expression of several proinflammatory cytokines and chemokines [38,39]. To our knowledge, no studies, to date, have investigated the potential mechanisms of CAP-associated performance improvements, specifically whether CAP may alter the inflammatory or endocrine responses to exercise and thereby influence the fatigue response in humans. 

Accordingly, given the paucity of data, we sought to explore the potential impact of acute oral CAP consumption on exercise performance, fatigue, and the inflammatory-endocrine response using a blinded, placebo-controlled, counterbalanced crossover design. The primary goal of our study was to better understand the intrinsic physiological effects of capsaicin administration in young, healthy individuals and to fill a gap in the literature concerning the ergogenic and fatigue-resistance of capsaicin in humans. To accomplish this, we employed the twitch interpolation technique to reveal the extent of peripheral fatigue and interpret the central nervous system contribution (voluntary activation) to the maximal voluntary contraction. We hypothesized that CAP supplementation would improve cycling performance and/or attenuate the observed neuromuscular fatigue after a cycling exercise time to exhaustion trial using the interpolated twitch technique, which might be due to an attenuated endocrine and inflammatory response to exercise.

## 2. Materials and Methods

### 2.1. Subjects and General Procedures 

Thirteen young and physically active males were recruited for this study from Skidmore College and the surrounding community. To be included, participants must have been healthy without any history of cardiovascular, neuromuscular, pulmonary, or metabolic diseases. Additionally, participants could not be current or recent (less than 6 months) smokers, have any known allergies and/or excessive sensitivity to spicy foods (i.e., hot peppers, jalapenos, paprika, etc.) or fiber (psyllium husk). Participants’ health history and eligibility were screened using health questionnaires to assess for eligibility (AHA/ACSM Pre-Participation Screening Questionnaire and Physical Activity Readiness Questionnaire [PAR-Q]). Participants were asked to refrain from consuming any vitamins or ergogenic supplements (i.e., L-Arginine, Citrulline-Malate, Pre-Workout) at least 2 days prior to each experimental visit, and to abstain from alcohol and caffeine, 24 h prior to testing. They were asked to report to the lab 2 h prior to the tests. All participants provided written informed consent prior to participation in the study. The study protocol was conducted in accordance with the most recent revisions of the Declaration of Helsinki and was approved by the Institutional Review Board (IRB#1807-733) and Institutional Biosafety Committees of Skidmore College.

### 2.2. Experimental Design

The subjects reported to the laboratory on three different days, with a minimum of 72 h between sessions (See Figure 1). Anthropometric and body composition data were collected on the first session using air displacement plethysmography (Bod Pod, Cosmed, Concord, CA, USA) [40]. Participants were then asked to perform a maximal incremental test on a magnetically braked cycle ergometer (828E, Monark, Cosmed, Vansbro, Sweden) starting at 50 W with increments of 25 W/min, at a self-selected cadence that was maintained for the duration of the incremental test as well as subsequent experimental trials. The test continued until participants were unable to continue the prescribed workload. At the end of the session, participants were familiarized with the isometric maximal voluntary contractions and the electrically evoked muscle contractions. In a single-blinded, counterbalanced, crossover design, on days 2 and 3, participants were asked to ingest either 2 × 390 mg of CAP capsules (Capsicool, Natures Way, Medley FL, USA) or 2 × 500 mg placebo pills (PL; Fiber, Psyllium Husk, Kirkland Signature, Seattle, WA, USA). The capsules were of similar appearance (e.g., color, size, etc.), taste (both were encased with cellulose/hypromellose capsules), and were coded inconspicuously to ensure blinding. The dosing was in accordance with manufacturer-suggested guidelines and was well-tolerated in pilot testing. The time to peak in serum concentration of CAP after oral ingestion is ~1 h [41]; for this reason, fatigue assessment at rest was assessed 50 min after pill ingestion to ensure adequate bioavailability. This was followed by a constant-load cycling exercise (85% of peak power output) to exhaustion (TTE) and another fatigue assessment immediate post-exercise (≤60 s). The neuromuscular assessment consisted of 6 maximal voluntary contractions (MVC) and superimposed twitch, pre and post the time to exhaustion trial. The bike test was terminated when the subjects could not maintain the self-selected pace for more than 10 s. Saliva samples were collected three times during experimental trials: before starting the first neuromuscular assessment, after the last neuromuscular assessment, and after 5 min of recovery.

### 2.3. Cardiorespiratory Exercise Responses

Ventilation (V_E_) and pulmonary gas exchange (VO_2_, VCO_2_) were measured breath-by-breath at rest and during the two trials through a mouthpiece and one-way non-rebreathe valve (Hans Rudolph 2700, Shawnee, KS, USA), nose clip, and the expiratory port coupled to a metabolic cart (TrueOne 2400, Parvomedics, Sandy, UT, USA) [42]. At the same time, central hemodynamic markers (HR: heart rate; SV: stroke volume; CO: cardiac output) were collected using a non-invasive thoracic impedance cardiograph (PhysioFlow^®^, Paris, France). The validity and reliability of this method have previously been established [43]. 

### 2.4. Assessment of Neuromuscular Function and Fatigue

The following methods were conducted in a manner similar to previous studies [44,45]. Accordingly, after proper skin preparation, two full-surface solid adhesive hydrogel-stimulating electrodes (size: 50 × 90 mm, Myotrode Plus, Globus G0465) were applied on the quadriceps: the anode was placed on the proximal part of the thigh, while the cathode was placed on the distal part of the leg extensors, 3 cm above the patella. The stimulation intensity was determined before the measurements by 25-mA increments until the size of the evoked twitch and compound muscle action potential (M-wave) demonstrated no further increase. The stimulated twitch force was measured by an adequately calibrated force transducer (MLP-300; Transducer Techniques, Temecula, CA, USA) statically connected to a custom-made chair through a non-compliant strap placed around the ankle of the self-reported dominant limb (right leg in all cases). The subjects were seated with a 90° knee flexion during the fatigue assessments. The superimposed twitch (SIT) and the resting twitch force (Q_tw,pot_) were measured during a 5-s MVC of the knee extensors and after 2-s of relaxed muscle. This procedure was repeated six times before and after the time to exhaustion cycling exercise. The data of the three best MVCs were analyzed and averaged. Voluntary muscle activation (VMA%) was calculated as VMA% = [1 − (SIT/Q_tw,pot_) × 100]. Peak force, maximal rate of force development (MRFD), and maximal relaxation rate (MRR) were analyzed for all Q_tw,pot_. Peak force was calculated as the highest value reached for every Q_tw,pot_, MRFD, and peak MRR of the resting twitch were calculated as the maximal steepness of the slope over a 10-ms interval. Data were collected using a Biopac system (MP150) and recorded using the AcqKnowledge A-D acquisition system (v. 4.4, Biopac, Goleta, CA, USA) on a separate computer. All data during the time to exhaustion were analyzed every 30 s. To understand the potential impact of CAP on perceptions of fatigue, we assessed the whole-body and leg rate of perceived exertion (RPE_tot_ and RPE_leg_, respectively) each minute during the trials. 

### 2.5. Microvascular Oxygenation 

Microvascular oxygenation was monitored with a multi-distance frequency-resolved near-infrared spectroscopy oximeter (NIRS; Oxiplex TS; ISS, Champaign, IL, USA). The NIRS technique provides non-invasive and continuous measurements of oxygenated (HbO_2_), deoxygenated (HHb), and total (Hbtot) hemoglobin levels, at a frequency of 2 Hz. The probe was calibrated each time prior to use and then positioned on the vastus lateralis of the non-dominant (left) leg, and secured with adhesive tape and a bandage to avoid light contamination, as in prior studies [46,47,48]. Due to identical spectral qualities, hemoglobin and myoglobin cannot be uniquely identified using NIRS, and thus represents a conglomerate signal.

### 2.6. Salivary Analysis

Samples of 1-mL whole saliva were collected as indicated above, via passive drool technique, and immediately stored at −80 °C until assay. Analysis of cortisol, IL-1β, IL-6, and α-amylase was conducted using commercially available ELISA and enzymatic kits (Salimetrics, Carlsbad, CA, USA). The assays were run with samples/standards in duplicate, in accordance with manufacturer guidelines, and read with a colorimetric spectrophotometer (iMark, Biorad, Hercules, CA, USA). The linearity for these assays was *R*^2^ > 0.99, while the coefficient of variation (CV) was <5% on standards for all assays.

### 2.7. Biochemical Analysis of Capsules

Capsaicin supplements (*n* = 3) and control fiber supplements (*n* = 3) were analyzed by extraction with ethanol to quantify the amount of the analytes capsaicin and dihydrocapsaicin in each supplement, as both act on TRPV_1_. The contents of each supplement were combined in 1.5 mL of ethanol and left to extract for eight hours in an oven at 80 °C with periodic shaking. The samples were filtered and the extract was analyzed by HPLC (Thermo Vanquish, Waltham, MA, USA) with mass spectrometric detection (Thermo ISQ-EC, Waltham, MA, USA) in order to quantify the capsaicin and dihydrocapsaicin content. External standards were used for calibration with a typical intra-assay CV of 3% and linearity of *R*^2^ > 0.995.

### 2.8. Statistical Analysis 

In a one-tailed, paired-sample design, an effect size of 0.8, and an alpha of 0.05, a sample size of 12 participants was estimated to ensure a statistical power of 0.80 (G*Power software, Kiel, Germany). Statistical comparisons were performed with commercially available software (Prism v. 8.0, GraphPad Software, San Diego, CA, USA). Data during the TTE (cardiovascular, ventilatory, inflammatory, and RPE variables) were analyzed using a two-way repeated-measures analysis of variance (ANOVA) to evaluate the differences between trials. Tests of normality and assumptions were conducted, if a significant violation was found, an appropriate adjustment to the degrees of freedom was made. For the TTE, the last time point was the subjective time to task failure. Paired samples t-tests were used to assess the differences between conditions in the pre-to-post TTE changes in the neuromuscular assessments. Statistical significance was declared when *p <* 0.05. Data are presented as Means ± SD, unless otherwise stated.

## 3. Results

### 3.1. Participant Characteristics 

Ten young, healthy, and physically active males met all inclusion criteria and completed all trials (Table 1). The pre-exercise cardiorespiratory parameters were not different between trials (all *p* > 0.05, data not shown). 

### 3.2. Supplement Analysis

Sample tracing of absorbance spectra for Capsaicin and dihydrocapsaicin used for subsequent quantification are presented in Figure 2. The average capsaicin content in each supplement was 0.957 mg/tablet with a range of 0.951–0.969 mg/capsule, thus the total dose was 1.914 mg. For dihydrocapsaicin, the average was 0.329 mg/capsule with a range of 0.326–0.332 mg/capsule, thus the total dose was 0.658 mg. The control fiber supplements contained no detectable levels of capsaicin or dihydrocapsaicin.

### 3.3. Exercise Performance, Neuromuscular Function and Fatigue

Both placebo and capsaicin conditions showed similar exhaustion (TTE) time of 375 ± 26 and 327 ± 36 s, respectively (*p* > 0.05, Figure 3A). Regarding the force before exercise, the MVCs were not different between the two conditions (640 ± 127 vs. 643 ± 161 N, *p* > 0.05), as well as after the TTE (479 ± 125 vs. 499 ± 133 N, *p* > 0.05). Accordingly, the baseline resting twitches (Q_tw,pot_) showed similar values (201± 64 vs. 205 ± 59 N, *p* > 0.05), but trended towards a greater Q_tw,pot_ immediately following exercise in the CAP condition as compared to the PL condition (100 ± 28 vs. 116 ± 37 N, *p* = 0.07, Figure 4F). This is also seen in the percentage change in post-exercise decline in Q_tw,pot_ in the two conditions, which reached statistical significance (−52 ± 6 vs. −42 ± 11 %, *p* = 0.037, Figure 4E). When the potentiated twitch (Q_tw,pot_ %) was plotted as a function of TTE, significant positive correlation with both PL (r = 0.7, *p* = 0.04) and CAP (r = 0.7, *p* = 0.04) was observed (Figure 3B). VMA% was not affected by either exercise or the supplement (*p* > 0.05). Looking at the intrinsic muscle contractile functions, MRR and MRFD showed significant reductions in pre-to-post TTE (*p* < 0.000). In addition, CAP mitigated the exercise-induced decline in MRR (*p* = 0.01; Figure 4C). Specifically, in the PL condition, MRR was reduced by 57 ± 22%, while only attenuated by 41 ± 19% in CAP. In contrast, MRFD decreased similarly in both conditions, namely, by 55 ± 16% and 49 ± 21% in PL and CAP, respectively (Figure 4D).

### 3.4. Microvascular Oxygenation during the TTE

Following CAP or PL ingestion, the pre-exercise levels of muscle oxygenation (StO2%; 64 ± 3 vs. 68 ± 8%), Total Hemoglobin Content (THC; 63 ± 23 vs. 66 ± 20 μM), Oxygenated Hemoglobin (HbO; 40 ± 14 vs. 44 ± 11 μM), and Deoxygenated Hemoglobin (Hb; 23 ± 10 vs. 22 ± 10 μM) were not different between conditions (*p* > 0.05). The start of the TTE modified the microvascular muscle oxygenation indexes, but the changes were not different with CAP treatment. However, the muscular circulation showed a general trend for higher values with CAP, which reversed during exercise, with THC (77.5 ± 28.1 vs. 80.2 ± 30.9 μM) and Hb (36.2 ± 20.3 vs. 40.2 ± 19.4 μM) higher in the PL condition. When we look at the hyperemia during recovery, CAP showed higher levels of StO2% as compared to PL (71.6 ± 1.6 vs. 69.5 ± 2.8%, *p* = 0.02), but there were no between-conditions differences for [THC] (90.1 ± 29.7 vs. 88.9 ± 31.8 μM), [HbO] (64.7 ± 22.0 vs. 62.3 ± 23.1 μM), and [Hb] (25.4 ± 7.9 vs. 26.7 ± 9.3 μM).

### 3.5. Central Hemodynamics, Ventilation and Perceived Exertion during the TTE

The indexes of the central hemodynamic (HR, SV, and CO) were not affected differently by the two conditions (Figure 5). No statistically significant condition x time interactions (*p* > 0.05) were observed for HR, SV, and CO during both baseline and exercise. As expected, there was a main effect of time for all the central hemodynamic markers (*p* < 0.00), but not an effect of CAP or PL administration. With CAP, the HR peak during exercise was 180 ± 7 bpm, SV was 212 ± 48 mL/min, and CO was 36 ± 8 L/min. In accordance, with PL the peak of HR was 181 ± 9 bpm, SV was 225 ± 49 mL/min, and CO was 38 ± 9 L/min. A significant time effect was found for the ventilatory responses to the exercise (*p* < 0.05) in VO_2_, VE, and RER (data not shown), whereas no interaction or condition effect was shown. Furthermore, the rating of perceived exertion (Figure 5D) of both whole-body and leg increased accordingly to the advance of the exercise and irrespective of the treatment (RPEtot: 7.8 ± 2.2 vs. 6.9 ± 2.8; RPEleg: 9.3 ± 1.3 vs. 9.0 ± 1.1; all *p* > 0.05). 

### 3.6. Stress and Proinflammatory Biomarkers

CAP did not influence salivary cortisol secretion at baseline, during, and after exercise. Indeed, there was a main effect of time (*p* = 0.002) with an increase in salivary cortisol concentration during recovery; however, CAP did not influence the overall kinetics (*p* > 0.05, Table 2). The activity of the salivary α-amylase tended to be reduced with CAP (*p* = 0.07), and in both conditions demonstrated a main effect of time (*p* < 0.001, Table 2). Looking at the interleukins, CAP increased the average salivary IL-6 concentration (*p* = 0.009) at baseline and during the exercise, then the level decreased to PL concentrations post-exercise. Moreover, CAP tended to attenuate the post-exercise rise in IL-1β (*p* = 0.053, Table 2).

## 4. Discussion

This study sought to determine the potential impact of acute oral capsaicin (CAP) administration on cycling endurance performance to exhaustion and parse out the associated physiological mechanisms underlying neuromuscular fatigue. Despite no differences between CAP and PL in cycling performance time to exhaustion, CAP did attenuate the post-exercise decline in the potentiated twitch. It partially impacted the contractile kinetics of the muscle, providing a greater rate of relaxation but no difference in the rate of contraction. CAP had no effects on the cardiorespiratory, perceptions of fatigue, or microvascular responses to the TTE trial. This suggests a potential enhancement of the sarcoendoplasmic reticulum Ca^2+^ ATPase (SERCA) pump activity, thereby preserving muscle relaxation. Moreover, CAP modulated changes in the pro-inflammatory interleukins, attenuating the rise of IL-1β during recovery. Partially in accordance with our hypothesis, CAP did not improve the time to exhaustion but it seemed to attenuate peripheral neuromuscular fatigue, increase muscle relaxation rate, and transiently alter the inflammatory response, independent of changes in cardiorespiratory or microvascular responses. 

### 4.1. CAP and Exercise Performance 

To date, just a few researchers have investigated the role of capsaicin during exercise in humans [28,29,30,31,32]. To our knowledge, this is the first study to investigate how CAP influences neuromuscular fatigue in humans in physiological terms and not just with perceptual indices. Indeed, acute CAP ingestion seems to increase performance or fatigue-resistance during a running time trial [30], high-intensity intermittent exercise [29], and resistance training [28]. However, in the present study, we did not observe any performance improvement (Figure 3), which agrees with the findings of Opheim and colleagues [32]. Previous work in rodents suggests that CAP increases performance in a dose-dependent manner [21,22,23,24], thus it is possible that the dose used in the present study was not enough to elicit a performance improvement; however, we might be the first to actually verify the capsaicin/dihydrocapsaicin content of the supplement and, importantly, we avoided any potential significant gastrointestinal distress that might have detracted from exercise performance. 

### 4.2. CAP and Exercise-Induced Neuromuscular Fatigue 

In the current study, after the cycling exercise trial, the indices of locomotor muscle fatigue were all negatively affected, as expected. Indeed, both the force and the voluntary muscle activation decreased to a similar extent regardless of CAP supplementation (Figure 4). Interestingly, the exercise-induced reductions in indices of peripheral fatigue were seemingly attenuated with CAP, chiefly amongst them, the maximal relaxation rate and the magnitude of the potentiated twitch. Mechanistically, a reason for these differences may be attributable to altered Ca^2+^ handling. It has been already documented that during intense exercise, the Ca^2+^ release from the sarcoplasmic reticulum (SR) is reduced in response to a marked depletion of cellular ATP [49,50], which may act to decrease the power output of the exercise and prevent peripheral fatigue from crossing a critical threshold [51]. This study investigated capsaicin, which increases the TRPV_1_ channel activity which can influence the Sarco/Endoplasmic Reticulum Calcium ATPase (SERCA) pump [52] in the muscle. Elevated SERCA pump activity with CAP-induced activation of TRPV_1_ in muscle consequently improved the kinetics of SR Ca^2+^ reuptake [49,50,53], perhaps explaining the better preserved maximal rate of relaxation post-exercise. Moreover, capsaicin may promote mitochondrial depolarization and reactive oxygen species (ROS) production, at least at high doses [52], but on the other hand is also purported to have remarkable antioxidant activity [54], particularly in lower dosages. Reactive oxygen species increase substantially during intense muscle activity [51] and are known to contribute to fatigue, but the relation between redox balance and performance is complex [55]. It is tenable that CAP could, in an antioxidant capacity, counteract the fatiguing effects of elevated ROS, perhaps better maintaining neuromuscular function post-exercise, but warrants further investigation. 

These outcomes highlight a potential role of CAP in attenuating the development of peripheral fatigue, perhaps via modulation of Ca^2+^ handling and its antioxidant effect. These findings are also supported by studies examining other antioxidants like ascorbate during exercise in healthy people [55,56] and in disease [57,58]. Moreover, if we integrate the microcirculation results, even if we see a tendency for higher StO_2_% and HbO during strenuous exercise, the significant rise in O_2_ delivery during recovery in CAP could improve peripheral vascular function [59]. The reason for no significant differences during exercise could be that CAP may affect muscle vasculature in higher doses than the ones we administered. However, in the present study, we aimed to minimize the potential side effects of capsaicin ingestion, namely gastrointestinal distress. We did not detect any differences in the indices of central fatigue, though previous studies in rats found that CAP activates subgroups of the metabosensitive group IV muscle receptors [60], whose stimulation reflexively increases the central drive [61]. Perceptually, it was previously found that acute CAP supplementation could decrease the rating of perceived exertion during endurance [28], though this was not the case in our study, as RPE increased equally during the time to exhaustion in both CAP and PL conditions.

### 4.3. CAP and the Physiological Response to Exercise

As capsaicin has been suggested to improve exercise performance and fatigue resistance, it is important to understand how it may alter the physiological response to exercise and ultimately support greater work. To this end, previous work in animal models suggests that CAP-induced improvements in performance were associated with increased hepatic glycogen content [21], perhaps due to glycogen sparing [24], and elevated fatty acid utilization as a result of catecholamine secretion and/or activity [22]. Moreover, a single high dose of CAP was found to downregulate the expression of the mitochondria uncoupling protein UCP3, which reduced the energy cost for a given electrically-induced contraction [25,27]. However, in humans, no change in muscle fat oxidation has been found with acute CAP supplementation during exercise recovery [62]. In the present study, the metabolic responses were similar throughout the exercise, suggesting that acute supplementation in humans does not affect energy expenditure, measured via VO_2_ (Figure 5), or energy substrate selection during exercise, as assessed by the RER, at least at this relatively high exercise intensity paradigm. Accordingly, the central hemodynamic and ventilatory responses were also similar between trials, in line with a similar metabolic cost. In addition, the microcirculation of the limb muscle also did not differ significantly during exercise, suggesting that CAP, at least in this dose, exerts minimal vasodilatory effect on the muscles. Indeed, during the initial rest, the muscular circulation showed a general trend for higher indices of microvascular perfusion with CAP, which reversed during exercise with THC and Hb higher with PL. Collectively, oxygen delivery and utilization seem unaffected by CAP supplementation and do not appear to be likely candidates for improved neuromuscular fatigue.

### 4.4. CAP and Neuroinflammatory Indices

In normal conditions, cortisol concentration after acute exercise is intensity-dependent [63] and increases to peak concentrations 20–30 min after the end of the physical activity [64]. Our results confirm the increasing cortisol trend after the end of the TTE, but CAP did not exert any effect on it. Indeed, it has been seen that repeated CAP administration in rats increases and prolongs the stress response [65], maybe to levels comparable to the ones of strenuous exercise, though this is typically seen with large doses. Looking at other salivary stressor biomarkers, CAP tended to lower the salivary α-amylase enzyme activity, perhaps indicative of lower sympathetic activity [66], perhaps via altered TRPV_1_ afferent activity. Although in vitro studies have demonstrated similar results, finding that capsaicin-derived compounds are potential α-amylase inhibitors [67], thus reinforcing our findings. Another important aspect is the anti-inflammatory properties of CAP. In our results, CAP attenuates the post-exercise rise in IL-1β, maybe blunting the proinflammatory cytokine production [39]. On the other hand, we found an increased salivary concentration of IL-6 after exercise that was unaffected by CAP [68,69], which could plausibly be the result of the strenuous performance [32,70], or the capsaicin-induced TRPV_1_ activation in adipose [71], or elsewhere. IL-6 may, in this case, have metabolic consequences [11,72] rather than inflammatory given the divergence between IL-6 and IL-1β. However, further work is needed in humans to decipher the potential impact of oral capsaicin on inflammation in humans and the potential ramifications on physiology and/or fatigue. Moreover, future research should look into larger and/or more chronic dosages of capsaicin and how they interact with lactate levels during exercise.

### 4.5. Limitations of the Study

This study was not conducted without limitation. First only young active males recruited from a college community were included, thus future work in older and/or female populations is needed. Second, the use of electrical stimulation on the muscle belly and not the femoral nerve may lead to lower neuromuscular responses. Lastly, more invasive measures of metabolism, including lactate and muscle level VO_2_, could be interesting to investigate during and after the exercise with CAP in future studies.

## 5. Conclusions

To our knowledge, this is the first study to investigate the effect of capsaicin on exercise performance, neuromuscular fatigue, and salivary indicators of stress and proinflammatory biomarkers in humans. Contrarily to the previous findings in humans, acute capsaicin administration did not improve exercise performance nor the rating of perceived exertion. However, it showed the capacity to attenuate peripheral fatigue development, which does not appear to result from changes in central hemodynamics, muscle oxygen delivery, or the magnitude of the central motor drive after the cycling exercise. Moreover, CAP modulated the saliva biomarkers, suggesting a potential depressed sympathetic activity and anti-inflammatory effect during the peak concentration with a late decrease in the proinflammatory markers. Collectively, capsaicin has the potential to alter the peripheral components of neuromuscular fatigue, leading to possible enhancements of exercise performance; the mechanisms underpinning such an observation have only begun to be understood in humans.

## Figures and Tables

**Figure 1 nutrients-14-00232-f001:**
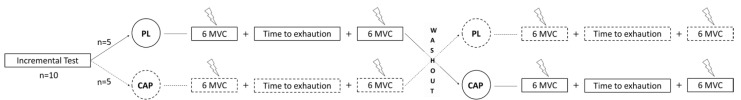
Experimental design of the study. After familiarization, participants reported to the lab on the first day for an incremental test. One week later, they were allocated in one of the two conditions, placebo (PL) or capsaicin (CAP). The neuromuscular assessment then started with 6 maximal voluntary contractions (MVC) using the interpolated twitch technique, which was repeated immediately following the time to exhaustion trial (85% W_peak_ of the incremental test). After one week of washout, the participants proceeded with the same assessments with the other condition.

**Figure 2 nutrients-14-00232-f002:**
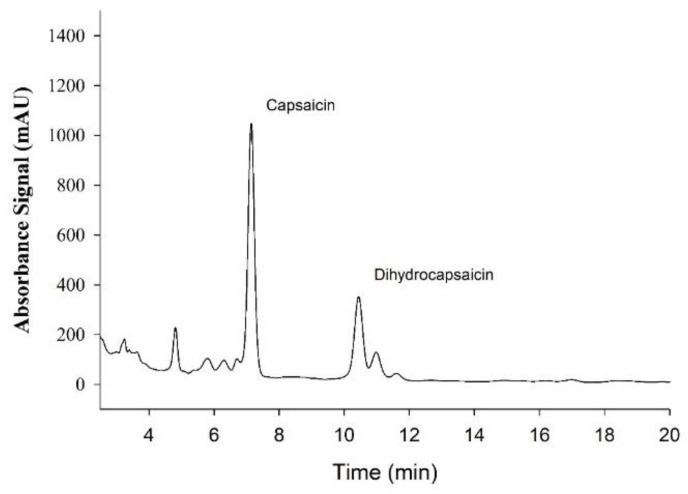
Sample Absorbance signal from High-Performance Liquid Chromatography (HPLC) analysis of Capsaicin supplement used for the quantification of the Capsaicinoids, Capsaicin, and Dihydrocapsaicin.

**Figure 3 nutrients-14-00232-f003:**
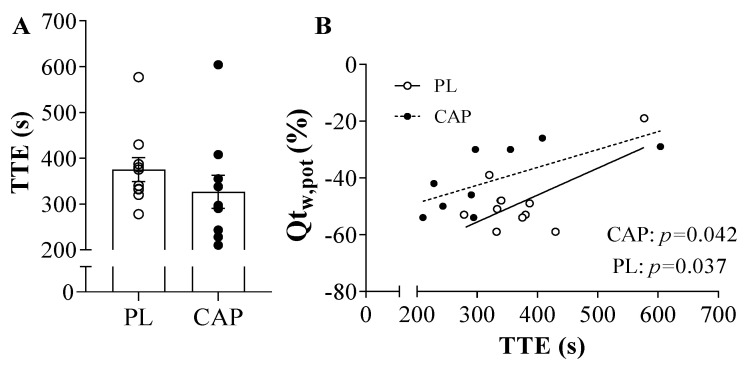
Time to exhaustion (TTE) and its correlation with the resting potentiated twitch (QT_w,pot_) after exercise (*n* = 10). (**A**) Time to exhaustion in individual valued after CAP or PL ingestion; (**B**) Time to exhaustion correlated with the QT_w,pot_ showed a significant positive correlation in both PL (*r* = 0.7, *p* = 0.04) and CAP (*r* = 0.7, *p* = 0.04). Values are presented as individual data and Mean ± SEM.

**Figure 4 nutrients-14-00232-f004:**
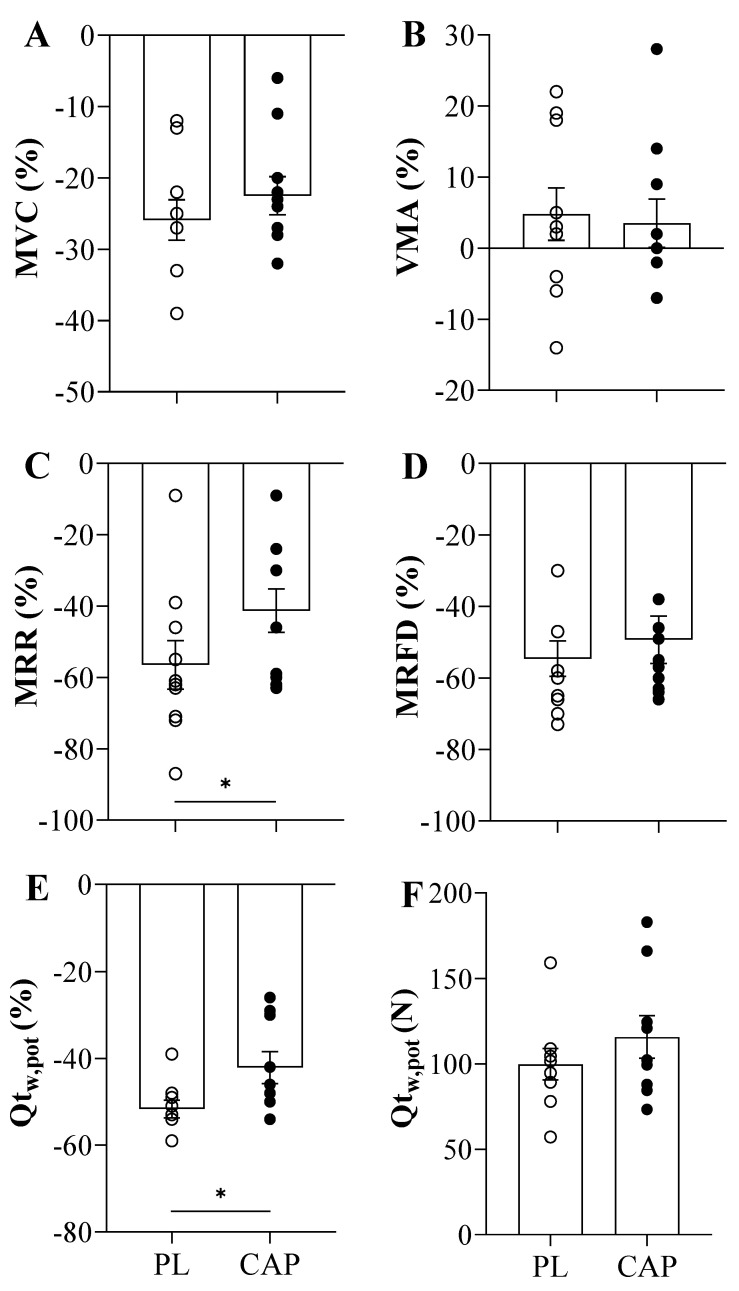
Neuromuscular Function Parameters expressed as the exercise-induced relative change after the time to exhaustion (TTE) in young active males (*n* = 10). (**A**) Maximal voluntary contraction. (**B**) Voluntary muscle activation. (**C**) Maximal relaxation rate. (**D**) Maximal rate of force development. (**E**) Resting potentiated twitch in percentual values. (**F**) Resting potentiated twitch in absolute values after fatigue. Values are presented as Mean ± SEM; *: *p* < 0.05.

**Figure 5 nutrients-14-00232-f005:**
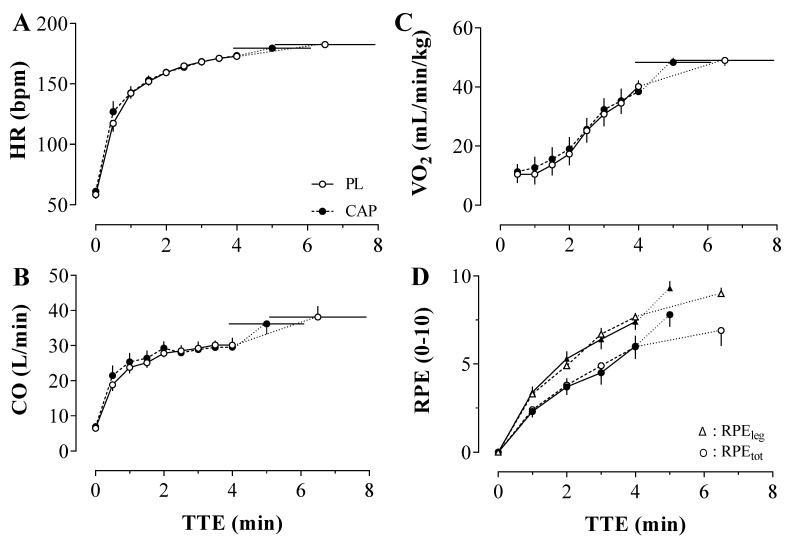
Central hemodynamic (Heart rate (HR); Cardiac output (CO); Panel (**A**,**B**)). Oxygen consumption (VO_2_; Panel (**C**)) and perceived exertion (RPE; Panel (**D**)) throughout the time to exhaustion (TTE) under Placebo (PL) and Capsaicin (CAP) Conditions (*n* = 10). In panel (**D**), RPE is represented as the rating of perceived exertion of the lower limbs (RPEleg; triangle) and whole-body (RPEtot; dot). Values are presented as Mean ± SEM.

**Table 1 nutrients-14-00232-t001:** Participant Characteristics.

Variable	Mean ± SD
Age (years)	22.3 ± 3.6
Height (cm)	182 ± 10
Weight (kg)	81.3 ± 11.5
Fat Mass (%)	11.7 ± 1.5
Peak Aerobic Power (Watts)	340 ± 21
Peak Oxygen Consumption (VO_2peak_, mL/kg/min)	49.5 ± 8.1

**Table 2 nutrients-14-00232-t002:** Endocrine and Inflammatory Biomarkers.

	Baseline		Exercise		10 min Post		15 min Post	
	PL	CAP	PL	CAP	PL	CAP	PL	CAP
**Cortisol (μg/dL)**	0.3 ± 0.2	0.3 ± 0.2	0.2 ± 0.0	0.3 ± 0.1	0.3 ± 0.2	0.3 ± 0.2	0.4 ± 0.2	0.4 ± 0.2
**α-amylase (U/mL)**	31 ± 36	15 ± 10	74 ± 41	57 ± 67	42 ± 29	40 ± 28	38 ± 27	22 ± 8
**IL-6 (pg/mL)**	10 ± 7	17 ± 14 *	7 ± 5	13 ± 7 *	8 ± 9	10 ± 7	8 ± 4	9 ± 5
**IL-1β (pg/mL)**	20 ± 18	21 ± 29	-	-	35 ± 40	15 ± 11	32 ± 28	15 ± 10 *

Data are presented as mean ± SD. IL-6: interleukin 6; IL-1β: interleukin 1β. *: *p* < 0.05 between PL and CAP—indicates low sample volume available for assay.

## Data Availability

The data presented in this study are available on request from the corresponding author. The data are not publicly available due to privacy and intellectual property concerns.

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
