# Peer review of "Capsaicin and Its Effect on Exercise Performance, Fatigue and Inflammation after Exercise"

_nutrients, 2022, doi:10.3390/nu14020232_

Round 1

Reviewer 1 Report

General comments:

Title

Are presented satisfactorily.

Abstract

Are presented satisfactorily. However, keywords are not found as descriptors in health sciences. please adapt.

Introduction

Are presented satisfactorily. However, the study problem must be better identified. There was no gap to justify the study.

Objectives must be presented better and more explicitly. Also, it would be good to present some hypotheses to be answered by the study.

Methods

It is not clear how the sample size was arrived at. Some calculation or statistical program was used to justify the sample size. please justify.

Results

Are presented satisfactorily.

Discussion

It should reaffirm the objectives and start discussing the results in the chronological order that appear in the item results. The limitations of the study were not presented.

There was no indication of future studies covering the gaps still left by the study.

Conclusion

Are presented satisfactorily. However, it should bring practical applications of the findings.

References

Of the 69 references, 17 are current and 52 have been published for more than five years. Please update the references and review the formatting of the references and text in view of the journal's standards.

Overview

The manuscript presented addresses a relevant research topic.

It would be advisable to do a general review.

Author Response

We would like to thank the reviewers for their time in examining our manuscript. We hope that the new version is satisfactory in all its parts. The changes are highlighted in red in the text.

 Reviewer#1

Title

Are presented satisfactorily.

RESPONSE: Thank you.

Abstract

Are presented satisfactorily. However, keywords are not found as descriptors in health sciences. please adapt.

RESPONSE: Thank you for the suggestion. The keywords have been adapted and re quite typically used in the health sciences.

Introduction

Are presented satisfactorily. However, the study problem must be better identified. There was no gap to justify the study.

RESPONSE: We would like to thank the reviewer for the suggestion. Although it was acknowledged as satisfactory, in agreement that the manuscript could be improved we have since updated the text to highlight the gaps in the literature (lines 58-60, 69-71, 78-80).

Objectives must be presented better and more explicitly. Also, it would be good to present some hypotheses to be answered by the study.

RESPONSE: We appreciate the input and regret if the study's purpose and premise were previously unclear. We have made significant revisions to the last paragraph of the introduction in response to this feedback.

Methods

It is not clear how the sample size was arrived at. Some calculation or statistical program was used to justify the sample size. please justify.

RESPONSE: We apologize that this was previously omitted, the sample size estimation procedure has since been added to the manuscript (lines 175-176). 

Results

Are presented satisfactorily.

RESPONSE: Thank you.

Discussion

It should reaffirm the objectives and start discussing the results in the chronological order that appear in the item results. The limitations of the study were not presented.

RESPONSE: Thank you. We think that this comment significantly increases the discussion section. The paragraph “CAP and the Physiological Response to Exercise” has been moved after the fatigue paragraph, to ensure a better alignment between the results and the discussion. Moreover, a limitation section has been included (lines 354-358).

There was no indication of future studies covering the gaps still left by the study.

RESPONSE: Thank you, two phrases have been added at the end of the discussion (lines 351-354).

Conclusion

Are presented satisfactorily. However, it should bring practical applications of the findings.

RESPONSE: Thank you, a phrase has been added at the end of the conclusion (lines 368-369).

References

Of the 69 references, 17 are current and 52 have been published for more than five years. Please update the references and review the formatting of the references and text in view of the journal's standards.

RESPONSE: Thank you for the suggestion. The authors made a supplementary literature research and found 3 papers published in the last 5 years that can suit in the manuscript (and are now added). However, the citations previous 2016 are important to introduce the topic, and a substitution is not appropriate.

Overview

The manuscript presented addresses a relevant research topic.

It would be advisable to do a general review.

RESPONSE: Thank you for recognizing the importance of this work and have made more broad revisions in response to this suggestion.

Reviewer 2 Report

Dear Authors.

Thank you for the opportunity to review your article under the title Capsaicin and its effect on exercise performance, fatigue, and inflammation after exercise. An article is well prepared and easy to follow. This research is very original as no studies to date have investigated the potential mechanisms of CAP-associated performance improvements, specifically potential alterations in neuromuscular fatigue and/or the inflammatory response in humans. The quality of presentation is fantastic and I would like to congratulate authors for such an amazing job done. There are only two requests that I have in regards with the manuscript in this form:

1) The authors inform that the sampled population represented 13 young and physically active males. It was not clear if this study is representative. What was the universe of the study? Did the sample procedures considered the total number of young and active males in the area? Could you, please, clarify these points?

2) What is the limitation of the study?

Once again, congratulations for a great job done.

Thank you!

Author Response

Reviewer #2

Dear Authors.

Thank you for the opportunity to review your article under the title Capsaicin and its effect on exercise performance, fatigue, and inflammation after exercise. An article is well prepared and easy to follow. This research is very original as no studies to date have investigated the potential mechanisms of CAP-associated performance improvements, specifically potential alterations in neuromuscular fatigue and/or the inflammatory response in humans. The quality of presentation is fantastic and I would like to congratulate authors for such an amazing job done.

RESPONSE: Thank you! We are extremely grateful for the positive review and constructive feedback.

There are only two requests that I have in regards with the manuscript in this form:

1) The authors inform that the sampled population represented 13 young and physically active males. It was not clear if this study is representative. What was the universe of the study? Did the sample procedures considered the total number of young and active males in the area? Could you, please, clarify these points?

RESPONSE: Thank you for the suggestion. The research was carried out in a collegiate setting with healthy young active individuals from the Skidmore College community and surrounding area. The average VO2 reflects their level of fitness (Table 1). This information has since been added.  Somewhat relatedly, we have also added information about the sample size estimation (lines 175-176).

2) What is the limitation of the study?

RESPONSE: A limitation section has been included at the end of the discussion (lines 354-358).

Once again, congratulations for a great job done.

Thank you!

RESPONSE: We appreciate the support and for such a collegial peer-review process.